# ACCO: Accumulate while you Communicate, Hiding Communications in Distributed LLM Training

## Abstract

Training Large Language Models (LLMs) relies heavily on distributed implementations, employing multiple GPUs to compute stochastic gradients on model replicas in parallel. However, synchronizing gradients in data parallel settings induces a communication overhead increasing with the number of distributed workers, impeding the efficiency gains of parallelization. To address this challenge, local optimization algorithms such as the ones used in Federated Learning have emerged. While effective in minimizing communication overhead, they incur significant memory costs, hindering scalability: in addition to extra momentum variables, optimizer's states cannot be partitioned among workers as communications are only allowed between rounds of local optimization steps. To conceal communication costs, we propose instead to synchronize delayed gradients *while* computing new ones between each model's update and introduce **AC**cumulate while **CO**mmunicate (`ACCO`), a memory-efficient optimization algorithm tailored for distributed training of LLMs. Accumulating local gradients on the workers until the communication finishes naturally reduces the idle time of GPUs and even allows the use of heterogeneous hardware. However, we show that the one-step delay inherent in parallel execution of gradient computations and communications has drastic impacts on Transformers' convergence. To compensate this delay we introduce a novel technique which leads to training dynamics aligned with standard distributed optimization. Compared to ZeRO, our implementation and experiments on several LLMs pre-training and fine-tuning tasks demonstrates that `ACCO` reduces the learning time up to 87% and successfully allows both sharding optimizer states across workers and the use of heterogeneous hardware.

## 1 Introduction

Training Large Language Models (LLMs) with billions of parameters requires thousands of GPUs running in parallel (Touvron et al., 2023). This relies on a distributed version of the backpropagation algorithm (Li et al., 2020) with a gradient-based optimizer such as Adam (Kingma & Ba, 2015) or AdamW (Loshchilov & Hutter, 2019). However at this scale, the communication overhead necessary to synchronize gradients between workers in the data parallel setting can dominate the time to compute the model updates (Ortiz et al., 2021), and it has been estimated that this will remain the case even if models and hardware evolve (Pati et al., 2023), hindering the benefits of parallelization. Moreover, as all workers are synchronized through gradient communication, the training only proceeds at the speed of the slowest machine (straggler) (Dutta et al., 2021; Mishchenko et al., 2022a).

To alleviate this issue, distributed optimization algorithms reducing the amount of communication between workers have been developed, such as local optimization methods (Stich, 2019; Wang et al., 2020b) which are especially used in Federated Learning (McMahan et al., 2017; Konecný et al., 2016). These methods authorize performing multiple optimization steps *locally* before communicating and synchronizing the distributed workers, reducing the communication overhead. As communication rounds can last longer than a local gradient computation (see Fig. 3), they also naturally allow to hide the cost of communications in the training time by running them in parallel to several consecutive local computation steps (Wang et al., 2020a; Shen et al., 2019; Zhang et al., 2015; Sun et al., 2024). Moreover, on heterogeneous hardware, the number of computation steps can be tuned

locally to the worker's speed so that slow ones compute less than fast ones, maxing out workers' usage (Diskin et al., 2021; Maranjyan et al., 2022).

However, this comes at a drastic memory cost. Indeed, in the standard data parallel setting, most of the memory consumption of model states comes from storing the optimizer's parameters, especially when training with mixed precision. To avoid the replication of redundant optimizer states across the workers, methods such as ZeRO (Rajbhandari et al., 2020a) shard them. Due to limited GPU memory and large models' size, all frameworks used in practice nowadays to train LLMs at scale use a form of partitioning method (Rasley et al., 2020; Andonian et al., 2023). However these sharding methods rely heavily on the fact that *each* mini-batch gradient is averaged over all the workers during the backward step. This is no longer the case with local optimization algorithms: if it were, then an averaging would happen at each step, defeating the purpose of the local method. This forces each worker to host a full copy of the optimizer's parameters, drastically increasing the memory requirements. Moreover, to prevent local steps from reducing the accuracy of the resulting model, local methods often introduce an outer optimizer step at each communication, which comes with additional momentum terms (Wang et al., 2020b; Sun et al., 2024). Hence, to store these variables, the latest state-of-the-art method CO2 (Sun et al., 2024) needs a memory overhead of 4 model copies compared to a standard distributed Adam, which itself uses an order of magnitude more memory than its sharded version (Rajbhandari et al., 2020a). This raises the following question:

*Is it possible to design a memory-efficient optimization algorithm that hides the communication cost of distributed training of LLMs and accommodates heterogeneous hardware?*

To hide the communication cost while being memory-efficient, making sharded optimizers compatible with the idea of overlapping gradient computations and communications seems natural. The concept of running two parallel processes is already present in the sharded optimization literature, but for a different purpose. ZeRO-Offload (Ren et al., 2021) introduces the "Delayed Parameter Update" (DPU) which allows running the optimizer on the CPU while computing and averaging gradients on the GPU. By running these processes in parallel, the gradients computed during one step are on a version of the model parameters that are no longer up to date, as they have been updated by the optimizer concurrently. In practice, this one-step staleness hurts convergence, and the method can only be used after sufficiently many warmup steps of non-delayed optimization (Ren et al., 2021).

**Contributions.** We introduce **AC**cumulate while **CO**mmunicate (`ACCO`), a memory-efficient optimization algorithm that **(1)** allows to shard the optimizer parameters across workers, **(2)** overlaps gradients computations and communications, hiding the communication overhead while **(3)** maximizing GPU usage, even with heterogeneous hardware. **(4)** We introduce a novel method to compensate for the one-step delay induced by parallel execution of the gradient computations and communications, removing the need for warmup steps and **(5)** perfectly matching the training dynamics of standard distributed optimization. Our experiments across multiple LLMs training and fine-tuning tasks consistently show that `ACCO` allows for significant time gains. **(6)** We will release an open-source parallel implementation of `ACCO` with the final version of the paper.

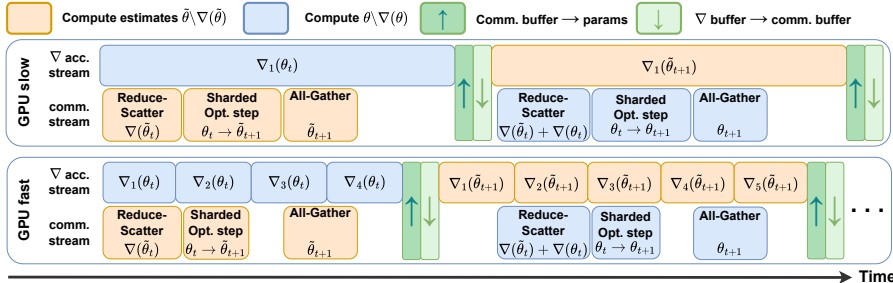

Figure 1: `ACCO` with a slow and a fast worker running in parallel, showing no idle time on both and hiding communications. The delayed update is compensated by splitting the mini-batch in two, leading to two steps in our timeline. The first uses half of the mini-batch to estimate "next step" parameters, and the second uses the full mini-batch to update them.

## 2 RELATED WORK

**Local optimization methods.** Local optimization methods perform several local model updates between periodic averaging. With the SGD optimizer, these algorithms predate the deep learning era (Zinkevich et al., 2010; McDonald et al., 2010), and their convergence properties are still investigated nowadays (Zhou & Cong, 2018; Stich, 2019; Woodworth et al., 2020; Mishchenko et al., 2022b). Due to their practical and efficient communication scheme, they have since been used for the Distributed Training of Deep Neural Networks (DNNs) with methods such as EASGD (Zhang et al., 2015), SlowMo (Wang et al., 2020b) or Post-local SGD (Lin et al., 2020; Ortiz et al., 2021), and are ubiquitous in Federated Learning (McMahan et al., 2017; Konecný et al., 2016; Li et al., 2019), broadening the choice of optimizers beyond SGD (Reddi et al., 2021; Karimireddy et al., 2020; Chen et al., 2020). By overlapping communications over consecutive steps of local computations, they allow to hide communication bottlenecks, resulting in algorithms such as Overlap local-SGD (Wang et al., 2020a), COCO-SGD (Shen et al., 2019) or CO2 (Sun et al., 2024). Moreover, with heterogeneous hardware, they can adapt their local computation rate to their hardware capacity (Diskin et al., 2021; Maranjyan et al., 2022). However this comes at the price of additional memory requirements: due to their local nature, not only do these methods prevent the use of sharded optimizers such as ZeRO (Rajbhandari et al., 2020a), but they also introduce additional control variables (Wang et al., 2020b; Mishchenko et al., 2022b; Sun et al., 2024), hindering their scalability as shown in Tab. 1. Moreover, catering for heterogeneous hardware is not straightforward, as using different numbers of local updates leads to models shifting at different speeds, requiring extra care to counter this effect (Maranjyan et al., 2022). On the contrary, ACCO does not lead to such disparities: it just affects *how* the required batch size is reached.

**Overlap decentralized optimization.** The communication complexity being a core concern in decentralized optimization (Yuan et al., 2016; Gorbunov et al., 2022), strategies have been devised to reduce communication overheads. For synchronous methods, works focus on designing algorithms with accelerated communication rates, leveraging Chebyshev polynomials (Scaman et al., 2017; Kovalev et al., 2020; Song et al., 2023). For the asynchronous ones, they rely on the properties of the graph resistance (Even et al., 2021; Nabli & Oyallon, 2023; Nabli et al., 2023). Alternatively, some approaches overlap gradient and communication steps, either explicitly (Assran et al., 2019), or by modeling them with independent stochastic processes (Nabli & Oyallon, 2023; Nabli et al., 2023). However, none of these works focus on memory efficiency. Thus, they introduce additional variables and do not consider sharding the optimizer states. Moreover, they do not study optimizers other than SGD, and extending their beneficial properties to adaptive methods commonly used for DNN training such as Adam is still an ongoing research topic (Assran et al., 2020).

**Memory-efficient distributed training of LLMs.** The activation memory overhead required for training Transformers (Vaswani et al., 2017) can be mitigated for an extra computational cost by reconstructing the input with reversible architectures (Jacobsen et al., 2018; Mangalam et al., 2022), or recomputing the activations via checkpointing (Chen et al., 2016). Efficient LLM training also combines parallelism methods. Classical data parallelism (DP) (Dean et al., 2012) suffers both from a high communication volume and a linear increase in memory due to the model replicas. ZeRO-DP (Rajbhandari et al., 2020b) and Fully-Sharded DP (Zhao et al., 2023b) avoid this issue by sharding the model states (i.e., the optimizer states, gradients, and parameters) between workers. This comes at the cost of further increasing the synchronization between workers and the communication volume, which can be mitigated by compression (Wang et al., 2023), memory trade-offs (Zhang et al., 2022), or delayed gradients (Fournier & Oyallon, 2024). The memory can be even more reduced using expensive CPU-GPU communications to unload states on the CPU (Ren et al., 2021; Rajbhandari et al., 2021). On the other hand, model parallelism partitions the DNN components for parallelization, either with tensor parallelism (Shoeybi et al., 2019) by slicing a layer's computation on several workers, or with pipeline parallelism, which divides a model into sets of layers trained in parallel on mini-batch slices. Popularized by Huang et al. (2019), this method leaves some workers idling and an inefficient memory overhead (Fan et al., 2021). Allowing delay in the gradients avoids worker idleness (Narayanan et al., 2019; Zhuang et al., 2020) but exacerbates the memory overhead, which can be partially mitigated with gradient accumulation (Narayanan et al., 2021; Zhuang et al., 2021) and activation checkpointing (Kim et al., 2020; Liu et al., 2023). Combining these frameworks results in the effective 3D parallelism (Smith et al., 2022).

**Delayed updates.** Delays being intrinsic to distributed asynchronous optimization, there is a rich literature studying them. In the case of distributed SGD in a parameter server setting, while early analysis showed convergence rates depending on the *maximal* delay (Agarwal & Duchi, 2011; Stich & Karimireddy, 2020b), recent lines of work improved these dependencies (Koloskova et al., 2024; Wu et al., 2022; Feyzmahdavian & Johansson, 2023), proving that asynchronous SGD beats standard mini-batch SGD even with unbounded delays (Mishchenko et al., 2022a). However, they only study plain SGD, which is hardly used for DNN training. In this context, some work focused on the interplay between SGD with momentum and delays (Mitliagkas et al., 2016; Zhang & Mitliagkas, 2019), while delay compensation schemes such as re-scaling updates (Zheng et al., 2017; Xie et al., 2020) or buffering them (Nguyen et al., 2022) were proposed for Federated Learning. But still, they only study versions of SGD and not adaptive methods commonly used for LLMs training such as Adam (Kingma & Ba, 2015) or AdamW (Loshchilov & Hutter, 2019). Closer to our work, DPU was introduced as a memory-efficient way to train LLMs by running the optimizer on the CPU while gradients are computed on the GPU (Ren et al., 2021), inducing a one-step delay between the gradients computed and the corresponding optimizer step. To mitigate it, they advise starting training by warming up for several steps with a standard method with no delay. Perhaps surprisingly, we find in our experiments that this one-step delay has a noticeable influence on the convergence of LLMs training, even when using warmup steps. Contrary to DPU, we remove the need for them, with no impact on the convergence of our training. Moreover, as it is not its purpose, DPU still runs communications in the gradient computation stream, and is thus impacted both by the communication overhead of scaling and hardware heterogeneity. Finally, in pipeline parallelism, gradient delays also affect computation, and simple weight prediction methods have been proposed to mitigate their effect (Chen et al., 2019; Yang et al., 2021). More elaborate predictions have been proposed for SGD to further reduce the impact of the delay (Kosson et al., 2021; Yang et al., 2020).

Table 1: Characteristics and memory consumption of several methods. $\Psi$: number of parameters in the model. $N$: number of workers. $K$: memory multiplier of the optimizer (Adam or AdamW). For SlowMo (Wang et al., 2020b) and CO2 (Sun et al., 2024), no mention of mixed precision training is made. We assume they use it and that their additional terms are stored in half precision. While no additional momentum is required for our method, we still need a communication buffer.

| Method | No comm. overhead | Handle hetero. hardware | Sharded Opt. | No add. momentum | Memory consumed per worker | $K = 12$, $N = 64$, $\Psi = 7.5B$ |
|---|---|---|---|---|---|---|
| Baseline DDP (Li et al., 2020) | ✗ | ✗ | ✗ | ✓ | $(2+2+K)\times\Psi$ | 120 GB |
| ZeRO-1 (Rajbhandari et al., 2020a) | ✗ | ✗ | ✓ | ✓ | $(2+2+\frac{K}{N})\times\Psi$ | 31 GB |
| SlowMo (Wang et al., 2020b) | ∼ | ✗ | ✗ | ✗ | $(2+2+2\times2+K)\times\Psi$ | 150 GB |
| CO2 (Sun et al., 2024) | ✓ | ✗ | ✗ | ✗ | $(2+2+4\times2+K)\times\Psi$ | 180 GB |
| ACCO (Ours) | ✓ | ✓ | ✓ | ✓ | $(2+2+2+\frac{K}{N})\times\Psi$ | 46 GB |

## 3 METHOD

In this section, we describe our method, including the approach to compensate for the delayed update. The algorithm will be described from the point of view of each worker $i \in \{1, ..., N\}$.

**Delayed Parameter Update.** First, we explain the presence of a delay by re-purposing the "Delayed Parameter Update" (DPU) (Ren et al., 2021) to fit in our framework. Contrary to the original DPU, we run gradient communications in the same stream as the optimizer step, in parallel to the gradient computations. To prevent GPU $i$ from being idle at step $t$, gradients are accumulated over as many mini-batches $N_i^{(t)} \geq 1$ as necessary until the communication process finishes, which varies depending on the speed of the worker as shown in Fig. 1. Each worker $i$ starts from the same neural network parameters $\theta^{(0)} \in \mathbb{R}^d$. $F : \mathbb{R}^d \to \mathbb{R}$ is the differentiable loss computed by our workers. A random mini-batch (modeled through the random variable $\xi \in \Xi$ following some law $\mathcal{P}$) is drawn from the local data shard $\mathcal{D}_i$ to initialize the stochastic gradient $g_i^{(-1)} = \nabla F(\theta^{(0)}, \xi_i^{(0)})$ and $N_i^{(-1)} = 1$. Then, for $t \in [\![0, T]\!]$ we repeat the following, the left and right sides running in parallel:

$$g_i^{(t)} = \sum_{k=1}^{N_i^{(t)}} \nabla F(\theta^{(t)}, \xi_{i,k}^{(t)}) \quad , \quad \theta^{(t+1)} = \texttt{Opt}\left(\theta^{(t)}, \frac{\sum_i g_i^{(t-1)}}{\sum_i N_i^{(t-1)}}\right) , \quad \text{(DPU)}$$

where `Opt` is the optimizer of our choice (*e.g.* Adam or AdamW for LLM training). Note that the right side combines both the gradient averaging (communications) and the optimizer step, which runs in parallel to the gradient computations to the left. Remark that, except at the first step $t = 0$, the gradients used by `Opt` are computed on parameters $\theta^{(t-1)}$ which differ from $\theta^{(t)}$, the ones we apply them to. This is inherently due to the parallel nature of our execution, and what we denote by "delayed update". We show in Sec. 5.2 that this has drastic impacts on the convergence in practice.

**Toward `ACCO`.** To counter this, we estimate what *would* be the parameters $\theta^{(t+2)}$ in addition to computing $\theta^{(t+1)}$. This allows the gradients at the next round to be computed on these estimates rather than the parameters of the last step. We denote this rule by "Weight Prediction" (WP). We initialize a common $\theta^{(0)}$, $\tilde{g}_i^{(0)} = \nabla F(\theta^{(0)}, \xi_i^{(0)})$, $N_i^{(0)} = 1$ and $\tilde{\theta}^{(1)} = \texttt{Est}(\bullet)$, where `Est` is our estimation function that could take any argument at this point. This leads to the following:

$$\tilde{g}_i^{(t+1)} = \sum_{k=1}^{N_i^{(t+1)}} \nabla F(\tilde{\theta}^{(t+1)}, \xi_{i,k}^{(t+1)}), \; \theta^{(t+1)} = \texttt{Opt}\left(\theta^{(t)}, \frac{\sum_i \tilde{g}_i^{(t)}}{\sum_i N_i^{(t)}}\right), \; \tilde{\theta}^{(t+2)} = \texttt{Est}(\bullet). \quad \text{(WP)}$$

Thanks to `Est`, the optimizer now applies to the parameters $\theta^{(t)}$ the gradients that were computed on an *estimated version* $\tilde{\theta}^{(t)}$, compensating the one-step delay. Akin to the idea of Chen et al. (2019) to counter delays in pipelining, a simple estimation function could be to re-use the gradients just received and apply a second optimizer step, *i.e.* using $\tilde{\theta}^{(t+2)} = \texttt{Opt}\left(\theta^{(t+1)}, \frac{\sum_i \tilde{g}_i^{(t)}}{\sum_i N_i^{(t)}}\right)$. We investigate this method (denoted by `ACCO`-wp) in Sec. 5.2, but found that its training dynamic differs from the baseline, whereas `ACCO`, the algorithm we present next, perfectly matches it. The

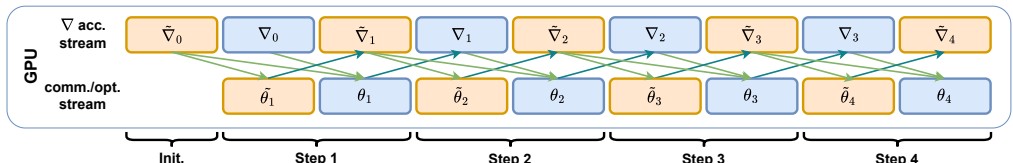

Figure 2: `ACCO`'s two-stage mechanism 1-2 to compensate the delayed updates.

crux of `ACCO` is to split the computation of the mini-batch gradients into two successive stages, where the first half of the mini-batch is used to estimate $\tilde{\theta}^{(t+1)}$ while $\theta^{(t+1)}$ is computed using the full mini-batch. This is motivated by the fact that gradient accumulation is often used to reach the extremely large batch sizes required to train LLMs (Zhao et al., 2023a), and if gradients are computed *sequentially* on a worker, we can leverage this to produce our estimate. Thus, starting with an initialized $\theta^{(0)}$, $\tilde{g}_i^{(0)} = \nabla F(\theta^{(0)}, \xi_i^{(0)})$ and $N_i^{(0)} = 1$, the two stages illustrated in Fig. 2 are (left and right side running in parallel):

$$g_i^{(t)} = \sum_{k=1}^{N_i^{(t)}} \nabla F(\theta^{(t)}, \xi_{i,k}^{(t)}) \qquad , \quad \tilde{\theta}^{(t+1)} = \texttt{Opt}\left(\theta^{(t)}, \frac{\sum_i \tilde{g}_i^{(t)}}{\sum_i \tilde{N}_i^{(t)}}\right), \quad (1)$$

$$\tilde{g}_i^{(t+1)} = \sum_{k=1}^{\tilde{N}_i^{(t)}} \nabla F(\tilde{\theta}^{(t+1)}, \tilde{\xi}_{i,k}^{(t+1)}) \quad , \quad \theta^{(t+1)} = \texttt{Opt}\left(\theta^{(t)}, \frac{\sum_i g_i^{(t)} + \tilde{g}_i^{(t)}}{\sum_i N_i^{(t)} + \tilde{N}_i^{(t)}}\right). \quad (2)$$

We describe the different components of our two-stage mechanism as follows:

1. The gradient computation stream uses the second half of the mini-batch to compute the gradients $g_i^{(t)}$ with respect to parameters $\theta^{(t)}$ while the communication stream estimates what would be the next steps parameters $\tilde{\theta}^{(t+1)}$ using the estimated gradients $\tilde{g}_i^{(t)}$.

2. The computation stream uses the first half of the mini-batch to estimate what would be the gradients $\tilde{g}_i^{(t+1)}$ of the next parameters $\theta^{(t+1)}$ using estimated parameters $\tilde{\theta}^{(t+1)}$ while the

communication stream computes $\theta^{(t+1)}$ using the full mini-batch. Note that it starts from the same version of the parameters $\theta^{(t)}$ as in step 1. The first half $\tilde{g}_i^{(t)}$ was estimated at step 2 of the *last round*, while the second half $g_i^{(t)}$ was just computed in 1.

**Theoretical discussion.** We can view DPU (with SGD as the optimizer `Opt`) as a parallel implementation of a Delayed-SGD (D-SGD) with a one-step delay. This algorithm with a delay of one has been studied in the convex setting, and is shown to converge at the same rate as SGD for quadratics (Arjevani et al., 2020) as well as for strongly and quasi convex functions (Stich & Karimireddy, 2020a). Thus, one could hope that it would generalize to adaptive optimizers and non-convex functions such as the ones met when training DNNs. However in practice, when training LLMs with AdamW, our experiments in Sec. 5.2 reveal that this one-step delay drastically hurts performances. To remove the impact of staleness, `ACCO` avoids using delayed gradients. Indeed, with SGD as optimizer and learning rate $\gamma > 0$, the parameter update of equation 2 reads

$$\theta^{(t+1)} = \theta^{(t)} - \gamma \sum_{i=1}^{N} \frac{\sum_{k=1}^{N_i^{(t)}} \nabla F(\theta^{(t)}, \xi_{i,k}^{(t)}) + \sum_{k=1}^{\tilde{N}_i^{(t)}} \nabla F(\tilde{\theta}^{(t)}, \tilde{\xi}_{i,k}^{(t)})}{N_i^{(t)} + \tilde{N}_i^{(t)}} .$$

This can be interpreted as a form of plain SGD with no delay, and a potentially variable batch-size (modeled through the $N_i^{(t)}, \tilde{N}_i^{(t)}$) split in two parts. While `ACCO` uses a mix of stochastic gradients $\nabla F(\theta^{(t)}), \nabla F(\tilde{\theta}^{(t)})$, they are not delayed compared to the parameters updated $\theta^{(t)}$ (see Fig. 2 for details). We verify experimentally this interpretation in Sec. 5 by showing that training LLMs with `ACCO` and standard distributed AdamW with the same batch-size leads to the same losses.

## 4 EMPIRICAL MOTIVATION AND CLUSTER SETTING

We empirically motivate the need for methods mitigating communication overhead in Distributed Data Parallel (DDP) (Li et al., 2020). Our goal is to illustrate that the time spent communicating gradients can quickly trump the one used for computing them when using DDP to train LLMs. For that, we measure the time necessary to perform a forward and backward pass on a Llama-2 model (Touvron et al., 2023) with 7B parameters hosted on a single GPU, using a batch size maxing out its memory. We compare this to the time necessary to compute an All-Reduce on those gradients with the NCCL backend, varying the number of distributed workers. On all the following, we experiment on our local cluster of NVIDIA A100-80GB GPUs with 8 GPUs per node and an Omni-PAth interconnection network at 100

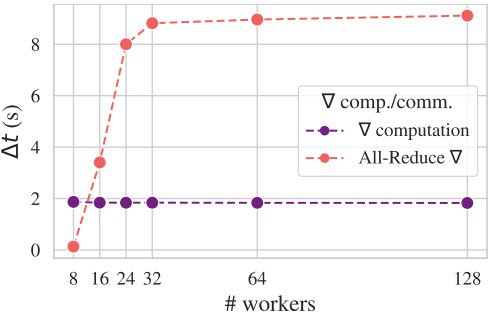

Figure 3: Time (per worker) spent computing and averaging gradients of a Llama-2 7B model for different numbers of GPUs.

Gb/s for inter-node connections, intra-node connections being done with NVLink 300 GB/s. Each distributed worker is hosted on a single GPU. We observe in Fig. 3 that when we communicate outside of a GPU node in our cluster, the time needed to average the gradients across workers can take more than *four times* the one spent on the whole forward and backward step. As DDP only partially hides communications during the backward (Li et al., 2020), this means that our GPUs remain idle the majority of the time when we use more than 24 distributed workers, motivating the need for methods leveraging this time to compute instead.

## 5 EXPERIMENTS

In this section, we lay down our experiments. First in Sec. 5.1, we detail the common setup for all our experiments. Second, in Sec. 5.2, we illustrate the failings of DPU and `ACCO`-wp that we hinted at in Sec. 3, which led us to crafting `ACCO`. For this first exploration, we focus on small language models and datasets, using TinyStories (Eldan & Li, 2023) as our test-bed. Then in Sec. 5.3, we verify that `ACCO` allows to efficiently train LLMs at scale by considering a 125M parameters GPT-Neo architecture (Black et al., 2021) and the OpenWebText dataset (Gokaslan et al., 2019).

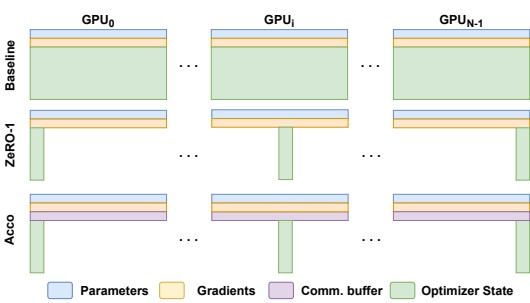

Figure 4: Memory requirements of ACCO vs DDP and ZeRO-1, see Tab.1 for quantitative details.

Finally in Sec. 5.4, we consider even larger models by using ACCO for an instruction fine-tuning task with a 2.7B parameters GPT-Neo, which accentuates the effects of the inter-node communication bottlenecks and highlights all the more the benefits of our method. They are further displayed in Sec. 5.5 where we compare between ACCO and DDP on heterogeneous hardware. Our method allows faster GPUs to accumulate while they wait for the slowest worker instead of remaining idle as in DDP, thus allowing us to compute gradients for large batch sizes faster than the baseline, resulting in quicker convergence in wall-clock time.

## 5.1 EXPERIMENTAL SETUP

All of our experiments are performed on the GPU cluster described in Sec. 4. A detailed pseudo-code for ACCO can be found in Appendix B.2. Our code is in Pytorch (Paszke et al., 2019), and we verified that our implementation produces two different CUDA streams running in parallel for the computations and communications using NVIDIA's Nsight System to profile it, as shown in Fig. 13. We trained all our models with AdamW (Loshchilov & Hutter, 2019), using mixed precision: our model parameters, gradient accumulation buffer, and communication buffers are in `bfloat16` (Kalamkar et al., 2019) while our sharded optimizer states are in single precision, as shown in Fig. 4. As nowadays all distributed frameworks training LLMs at scale use a form of partitioning due to GPU memory constraints (Rasley et al., 2020; Andonian et al., 2023), our main baseline is Pytorch's Distributed Data Parallel (DDP) (Li et al., 2020) with ZeRO-1 (Rajbhandari et al., 2020a) to shard the optimizer's state. As justified in Tab. 1, local optimization methods cannot be realistically considered for memory reasons. To compare in good faith DPU to ACCO in terms of wall-clock time, we also implemented our own version of DPU, as the available implementation (Ren et al., 2022) solves a different problem than ours. The original algorithm does not run parallel computation and communications as it is designed to host the optimizer on the CPU, and is slower than ZeRO due to recurrent memory transfers between CPU and GPU (Ren et al., 2021).

## 5.2 CRAFTING ACCO ON TINYSTORIES

Here, we experiment with small language models on the TinyStories dataset (Eldan & Li, 2023), following the configuration and training hyper-parameters of their paper (Eldan & Li, 2023) to the best of our abilities. Hence, we use a 36M parameters GPT-Neo based (Black et al., 2021) decoder-only transformer architecture. To match the 10k vocabulary they used, we trained our own BPE tokenizer on the TinyStories dataset. For our experiments, we used 8 workers on a single node.

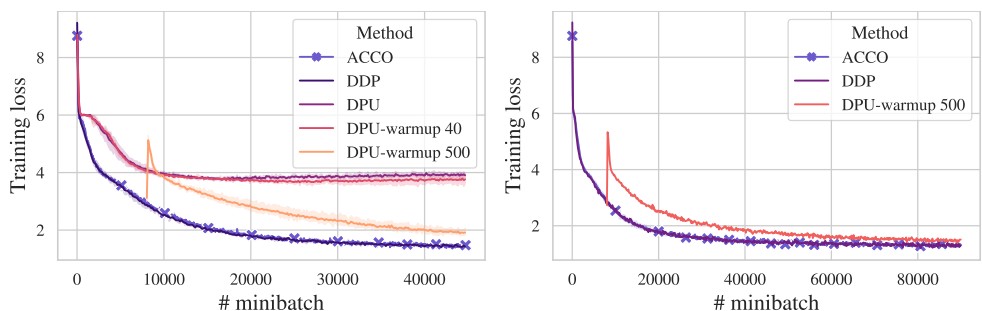

(a) Training with the specified amount in (Eldan & Li, 2023).

(b) Training for twice the specified amount.

Figure 5: Impact of the delayed update and the amount of warmup steps on the training

**Impact of delayed updates.** First, we investigate the impact of using delayed updates, re-purposing DPU (Ren et al., 2021) as described in Sec. 3. We run three variants of this algorithm: **(1)** with no warmup, **(2)** with 40 warmup steps of non-delayed optimization step before switching to DPU (recommended recipe in (Ren et al., 2021)), and **(3)** with 500 steps of warmup. We report in Fig. 5 our training losses on 8 distributed workers averaged over 3 runs. We remark that using delayed updates greatly hurts convergence, especially when no or too few warmup steps are performed. Surprisingly, the number of warmup steps given in (Ren et al., 2021) does not work here, hinting that it is a sensitive hyper-parameter to tune for each use-case. If we train for twice as long than specified in Eldan & Li (2023), then the DPU training curve approaches the baseline one, without out totally catching it. Contrary to this, the training curve of our algorithm `ACCO` perfectly matches DDP's one from the beginning.

**A simple approach to compensate delays.** To mitigate the detrimental impact of using delayed updates, we test a first approach to mitigate it: `ACCO-wp`, the Weight Prediction method described in Sec. 3. This method applies two consecutive optimizer steps, re-using the same mini-batch of gradients twice. The first step produces the usual updated parameters, while the second predicts the parameters of the next step so that gradients can be computed on this estimate rather than on a stale version of the model. In Fig. 6 we compare the training curves of this delay-compensation method to ours. We remark that, while `ACCO` perfectly matches the DDP baseline at all times, `ACCO-wp` displays worse behavior, especially at the beginning of the training. Thus, we dismiss this method and keep ours for the remaining of the experiments.

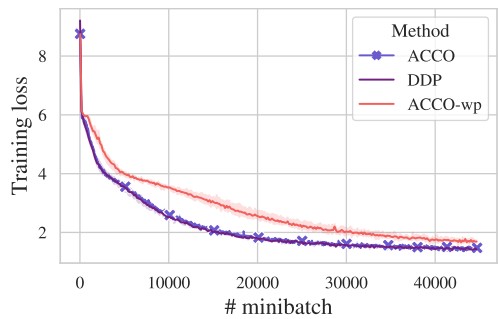

Figure 6: Comparison of `ACCO` with its Weight Prediction version on TinyStories.

### 5.3 Passing the scaling test: training GPT-Neo on OpenWebText

To assess how `ACCO` scales with larger models and more data, we pre-trained a model equivalent to GPT-2 (Radford et al., 2019) with both `ACCO` and DDP with a ZeRO optimizer. Specifically, we used the GPT-Neo architecture (Black et al., 2021) with 125 million parameters and the OpenWebText dataset (Gokaslan et al., 2019), which contains 40 GB of text. We used the GPT-Neo tokenizer, pre-trained on the Pile dataset (Gao et al., 2020). The models were trained on sequences of 1024 tokens, with documents concatenated using end-of-sequence tokens. To assess the impact of using different hardware, the experiment was repeated on 2 different clusters. The first was conducted on 8 H100-PCIe 80GB on a single node. The second was on 32 A100-80G GPU distributed on 4 nodes. We maxed out the memory of our GPUs with a local mini-batch size of 24. To reach a sufficiently large overall batch size, we used 1 step of gradient accumulation for DDP, and none for `ACCO` as our method naturally accumulates over 1 step, resulting for the first and second experiments in respectively 400K and 1.5M tokens per effective batch for both `ACCO` and DDP. In Tab. 3, we report additional experimental details, and notice that training with `ACCO` allows for a 25% speedup on this pre-training task, which is additionally illustrated in Fig. 7. We also report that our implementation of `ACCO` adaptively scheduled 315 supplementary accumulation steps over the whole training to prevent GPUs from idling while waiting for communications. Further details and results for the H100 experiment can be found in Appendix A. Tab. 2 reports the perplexity of trained language models with both methods. We evaluate the perplexity of language models on LAMBADA (Paperno et al., 2016) and a test split of OpenWebText, and report similar results for both methods.

Table 2: Perplexity of our trained LLMs

| Method | LAMBADA (ppl ↓) | OpenWebText (ppl ↓) |
|---|---|---|
| `ACCO` 1x8 | 47.1 | 24.2 |
| DDP 1x8 | 47.5 | 24.3 |
| `ACCO` 4x8 | 45.5 | 22.5 |
| DDP 4x8 | 44.1 | 21.7 |

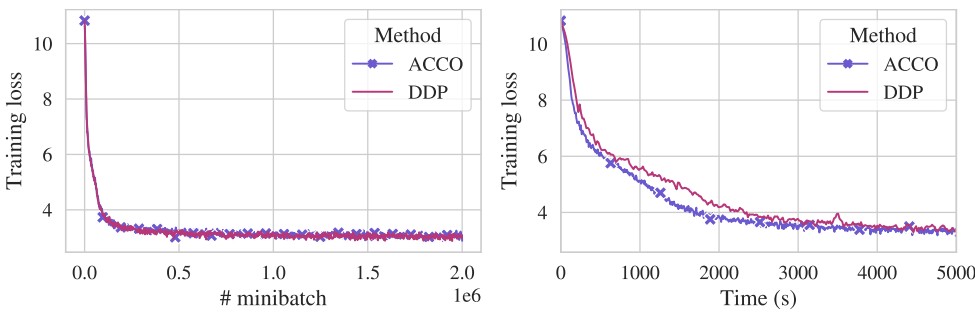

(a) Evolution of the loss over the whole training.   (b) Focus on the first part of the training w.r.t time.

Figure 7: Training curves for ACCO and DDP with 32 workers trained for 50B tokens.

## 5.4 ADVANTAGES OF USING ACCO FOR INSTRUCTION FINE-TUNING

In previous sections, we compared ACCO against DDP with ZeRO in the pre-training stage. To further validate our algorithm, we consider the GPT-Neo 2.7B model (Black et al., 2021) pre-trained on the Pile dataset (Gao et al., 2020) and finetuned it on the Alpaca dataset (Taori et al., 2023) containing 52k pairs of instruction/answer. We fine-tuned the model using two configurations: 8 A100-80G on a single node, and 8 A100-80G distributed equally across 2 nodes. Samples are padded to match the longest sequence in the mini-batch. We fixed the mini-batch size at 4, leading to a total batch size of 128 for all methods. For DDP and DPU, we used a gradient accumulation of 4, while for ACCO , a gradient accumulation of 2 to account for the ACCO accumulation described in Sec. 1. The learning rate was set to $2 \times 10^{-5}$ for all methods with a warmup of 50 steps, for DPU.

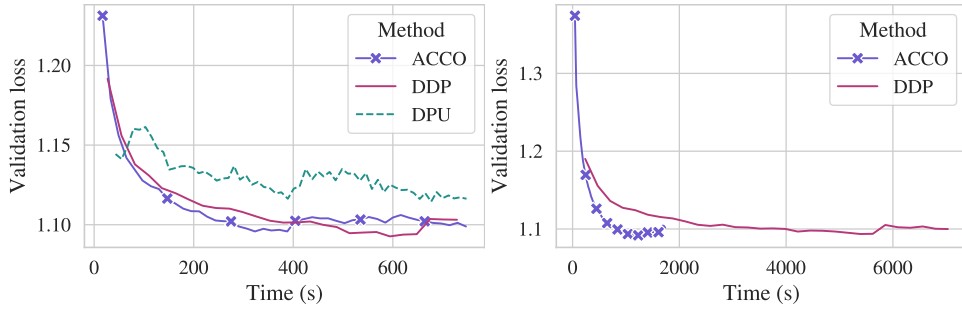

Figure 8: Validation curve with 8 workers on 1 node **(left)**, and 4 workers/node on 2 nodes **(right)**.

In this setting, padding to the longest sequence in the mini-batch induces more variability in the number of tokens per mini-batch. This results in more variability in the computational load for each worker, leading to increased wait times for synchronization. We observe in Fig. 8 that ACCO hits a low validation loss faster than DDP on both settings. Note that the difference between ACCO and DDP is accentuated when workers are distributed on multiple nodes, leading to a 87% speedup for ACCO (see Tab. 3) and highlighting the impact of communication bottlenecks on standard methods.

Table 3: Pre-training and finetuning time speedup with ACCO against DDP on various setups.

| Stage | Model | GPUs | #tokens | DDP w/ ZeRO-1 | ACCO | $(\mathbf{\Delta T})$ |
|---|---|---|---|---|---|---|
| **Pre-training** | GPT-Neo-125M | 1x8 | 6B | 4h41min | 4h25min | $(-5.69\%)$ |
| | | 4x8 | 50B | 14h41min | 10h55min | $(-25.65\%)$ |
| **Finetuning** | GPT-Neo-2.7B | 1x8 | 80M | 43min | 25min | $(-41.86\%)$ |
| | | 2x4 | 80M | 3h46min | 29min | $(-87.17\%)$ |

## 5.5 Experiment Using Heterogeneous Devices

To witness the impact of using heterogeneous devices, we run `ACCO` and compare it to DDP in a four workers setting, with one of the GPU four times slower than the other three. The training setting is the same as in Sec. 5.2. As we experiment on a A100 GPUs cluster, we simulate the heterogeneity of the hardware using the `time.sleep()` python command. First, we measure the time that a standard forward-backward step takes, and make one of the four GPUs idle for three times this amount after each forward-backward pass. In this context, DDP is only as fast as the slowest worker: 3 out of the 4 workers are idle 3/4 of the time. With `ACCO`, the other workers accumulate during the time they are waiting for the slow one to finish. Thus, `ACCO` allows to compute gradients for large batch sizes faster than standard baselines, resulting in faster convergence in terms of wall-clock time, as displayed in Fig. 9.

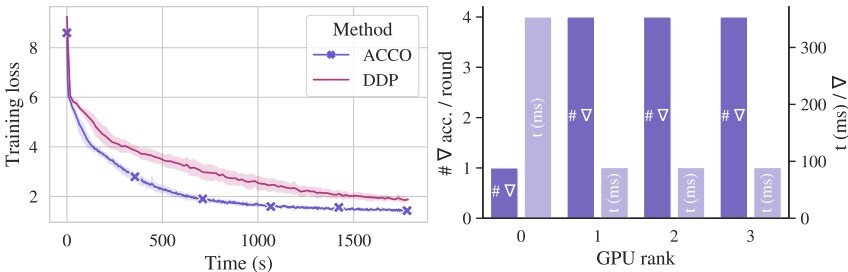

Figure 9: Training curves with 3 normal workers and 1 slow worker ($4\times$ slower).

## 6 Limitations

**Experiments mainly on one cluster environment.** Due to the lack of variety in the compute environments we have access to, the majority of our experiments were performed on a single cluster, described in Sec. 4. This is a communication-constrained setting, as our hardware is not the most cutting-edge in that regard as discussed in Sec. 4. However, to mitigate this one-sidedness, we also run a small pre-training study on one of the fastest hardware available today, and report in Tab. 3 that even in that case, `ACCO` leads to a 5% time gain.

**Communication cost only *hidden*, not reduced.** While local optimization methods tackle the communication overhead problem with scarce communications, here we only hide them. Thus, our method does not lead to energy savings, nor question the cost of highly synchronized infrastructure. However, `ACCO` naturally maximizes the hardware throughput, allowing to reduce their use time.

**Further memory savings avenue not explored.** Due to the parallel nature of `ACCO`, removing the reliance on communication and gradient buffers seems hardly possible, questioning the feasibility of further memory savings if all executions are kept on the GPU. But, akin to ZeRO-Offload (Ren et al., 2021), the communication and optimizer stream could entirely be run on CPU, which would allow significant memory gains. We did not experiment with this idea, and let it for future work.

## Conclusion

We propose `ACCO`, a novel algorithm that jointly addresses the memory and communication challenges inherent in training LLMs on distributed systems. By allowing for parallel computation and communication of gradients while partitioning the optimizer states, `ACCO` effectively reduces communication overhead in a memory-efficient fashion. We introduce a novel two-stage mechanism to compensate for the delayed update inherent to this parallel setting, which ensures consistent convergence dynamics with the standard optimization algorithm for large-scale distributed LLM training without the need for warmup steps. We empirically confirm the benefits of our methods over several pre-training and finetuning tasks, reporting drastically reduced training times compared to our baseline, especially in multi-node settings or with heterogeneous devices.

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

## A    EXPERIMENTAL DETAILS AND FURTHER RESULTS

### A.1    PRE-TRAINING ON TINYSTORIES

For experiments in Sec. 5.2, we used the configuration available on the Huggingface Hub [1]. We trained our own 10k vocabulary tokenizer on the dataset. We also report in Fig. 10 the results of our study on the impact of halving the batch size for DPU by not performing any gradient accumulation (*i.e.*, performing an optimizer's step at each communication).

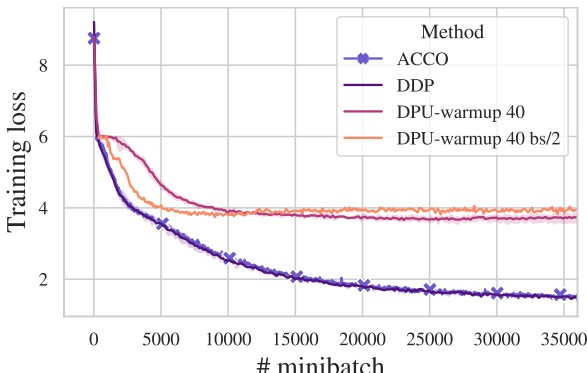

Figure 10: Comparison between running DPU on 8 GPUs with 2 steps of gradient accumulation on each (to match the standard batch-size) and DPU with only 1 gradient accumulation step. Doing so allows to double the number of optimizer's step per minibatch, but divides the effective batch size by 2. This leads to faster convergence early in the training, but worse training loss in the end.

### A.2    PRE-TRAINING ON OPENWEBTEXT

For all pre-training experiments on OpenWebText, the configuration used to instantiate the GPTNeo 125M is available on the Huggingface Hub[2]. We only changed the "max_position_embeddings" parameter from 2048 to 1024. More details are displayed in Tab. 4. We used the OpenWebText dataset available on Huggingface[3]. We also report in Fig. 11 further results for the pre-training on H100 GPUs.

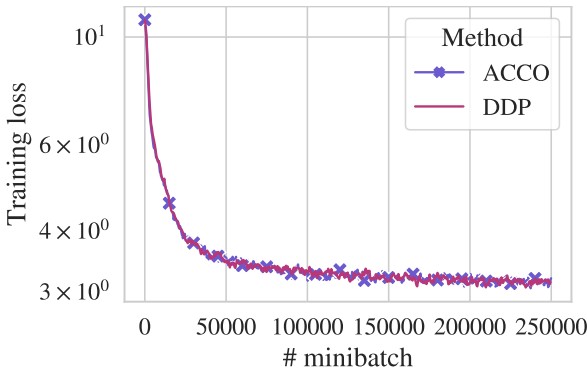

Figure 11: Training loss during training on OpenWebText with 8 H100 GPUs and 6B tokens.

---

[1]Tiny Stories Available at: https://huggingface.co/datasets/roneneldan/TinyStories

[2]GPT-neo 125M Configuration Available at: https://huggingface.co/EleutherAI/gpt-neo-125m/blob/main/config.json

[3]OpenWebText Dataset Available at: https://huggingface.co/datasets/Skylion007/openwebtext

Table 4: Training hyperparameters for ACCO and DDP configurations.

| Hyperparameter | 8 H100 | 32 A100 |
|---|---|---|
| mini-batch_size | 24 | 24 |
| n_grad_accumulation | ACCO: -DDP: 1 | ACCO: -DDP: 1 |
| sequence_len | 1024 | 1024 |
| #tokens_batch | 400K | 1.5M |
| optimizer | AdamW | AdamW |
| learning_rate | 6e-4 | 6e-4 |
| weight_decay | 0.1 | 0.1 |
| adam_beta1 | 0.9 | 0.9 |
| adam_beta2 | 0.95 | 0.95 |
| nb_steps_tot | 50000 | 50000 |
| scheduler | cosine | cosine |
| n_warmup_steps | 0 | 0 |

## A.3 INSTRUCTION FINE-TUNING

For all fine-tuning experiments, we used the pre-trained GPT-neo 2.7B available on the Huggingface Hub[4] and the associated tokenizer. We used the Alpaca dataset available on Huggingface[5]. More details are displayed in Tab. 5.We also report in Fig. 12 further results on the experiment described in Sec. 5.4.

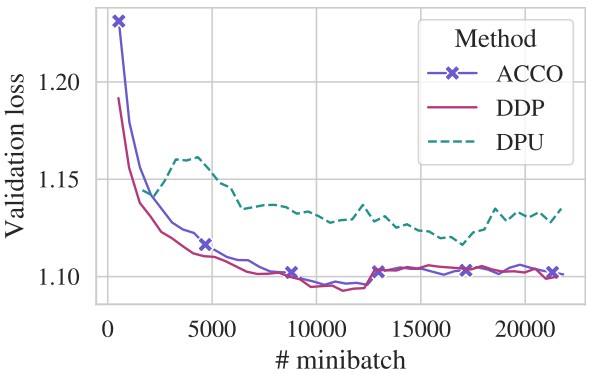

Figure 12: Validation curve with 8 workers on a single node.

Table 5: Finetuning hyperparameters for ACCO, DDP and DPU configurations.

| Hyperparameter | ACCO | DDP | DPU |
|---|---|---|---|
| mini-batch_size | 4 | 4 | 4 |
| n_grad_accumulation | 2 | 4 | 4 |
| total batch_size | 128 | 128 | 128 |
| optimizer | AdamW | AdamW | AdamW |
| learning_rate | 2e-5 | 2e-5 | 2e-5 |
| weight_decay | 0.0 | 0.0 | 0.0 |
| adam_beta1 | 0.9 | 0.9 | 0.9 |
| adam_beta2 | 0.95 | 0.95 | 0.95 |
| nb_steps_tot | 50000 | 50000 | 50000 |
| scheduler | cosine | cosine | cosine |
| n_warmup_steps | 0 | 0 | 50 |

---

[4]GPT-neo 2.7B Available at: `https://huggingface.co/EleutherAI/gpt-neo-2.7B`
[5]Alpaca Dataset Available at: `https://huggingface.co/datasets/tatsu-lab/alpaca`

# B    IMPLEMENTATION DETAILS

## B.1    PROFILING RESULTS

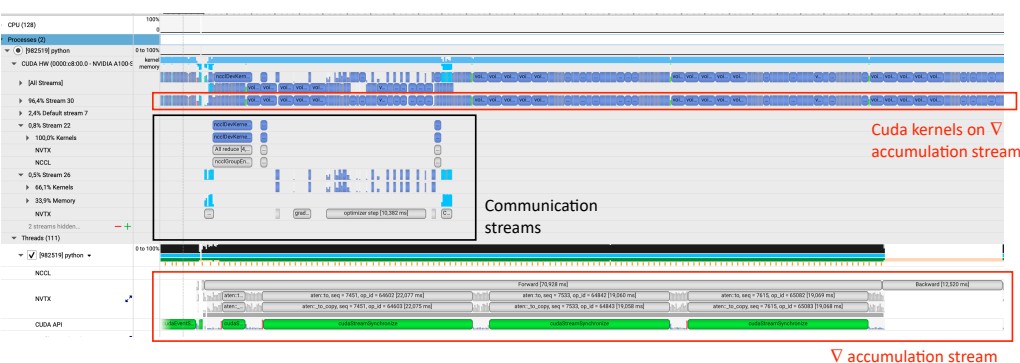

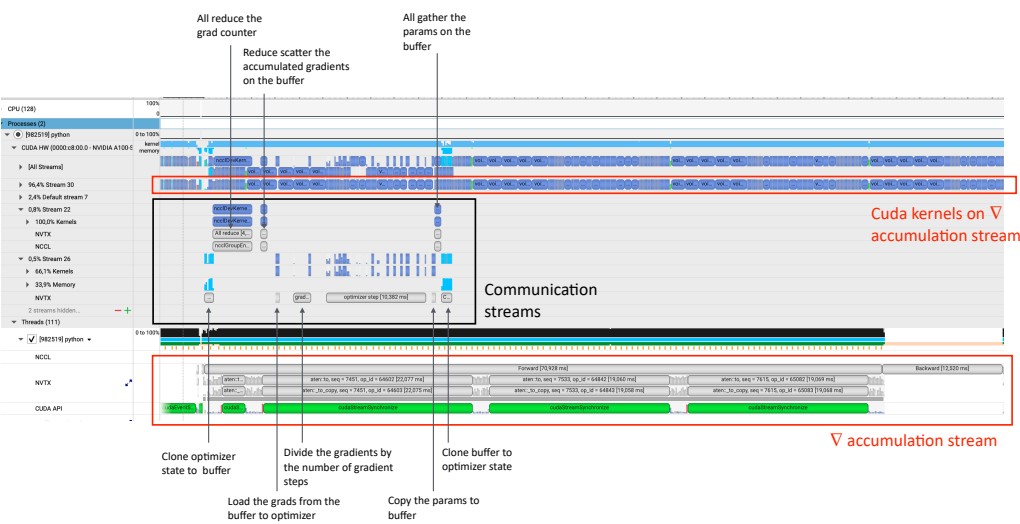

Figure 13: Nsight system profile of our implementation of `ACCO`: our two steams do run in parallel. In this Figure, the computation take more time than the communication because we only profiled small scale experiments with 8 workers, and small number of parameters (36M as we profiled on our TinyStories Eldan & Li (2023) setting). This changes when using larger models and more workers, as seen in 4.

## B.2    ALGORITHM PSEUDO-CODE

---

**Algorithm 1** Training with `ACCO` in parallel for a worker $i$

---

1: **Input:** Model with differentiable loss $F$, number of models $N$, initial parameters $\theta^{(0)}$, training steps $T$, dataset shards $\mathcal{D}_i$.

2: **Initialize:** gradients $g_i{}^{(-1)} = \nabla F(\theta^{(0)}, \xi_i^{(0)})$ and number of gradients $N_i^{(-1)} = 1$

3: **# Computation CUDA stream**

4: **while** $t < T$ **do**

5:     **Stage 1.**

6:     **while** not `Ready_for_Stage_2` **do**

7:         $\xi_i^{(t)} \leftarrow \mathcal{D}_i$

8:         $g_i^{(t)} \leftarrow g_i^{(t)} + \nabla F(\theta^{(t)}, \xi_i^{(t)})$

9:         $N_i^{(t)} \leftarrow N_i^{(t)} + 1$

10:     $\tilde{\theta}^{(t+1)} \leftarrow$ **Buffer**$_i$

11:     **Buffer**$_i \leftarrow (N_i^{(t)}, g_i^{(t)})$

12:     **Stage 2.**

13:     **while** not `Ready_for_Stage_1` **do**

14:         $\xi_i^{(t)} \leftarrow \mathcal{D}_i$

15:         $\tilde{g}_i^{(t)} \leftarrow \tilde{g}_i^{(t)} + \nabla F(\tilde{\theta}^{(t+1)}, \xi_i^{(t)})$

16:         $\tilde{N}_i^{(t)} \leftarrow \tilde{N}_i^{(t)} + 1$

17:         $t \leftarrow t + 1$

18:     $\theta^{(t+1)} \leftarrow$ **Buffer**$_i$

19:     **Buffer**$_i \leftarrow (\tilde{N}_i^{(t)}, \tilde{g}_i^{(t)})$

20:

21: **# Communication CUDA stream**

22: **while True do**

23:     **Stage 1.**

24:     $(\tilde{N}_i^{(t)}, \tilde{g}_i^{(t)}) \leftarrow$ **Buffer**$_i$

25:     $\sum_i \tilde{N}_i^{(t)} \leftarrow$ `All_Reduce`$(\tilde{N}_i^{(t)})$

26:     `Shard`$_i \left( \sum_i g_i^{(t)} \right) \leftarrow$ `Reduce_Scatter`$(\tilde{g}_i^{(t)})$

27:     `Shard`$_i \left( \tilde{\theta}^{(t+1)} \right) \leftarrow$ `ShardedOpt` $\left( \text{Shard}_i \left( \theta^{(t)} \right), \text{Shard}_i \left( \frac{\sum_i \tilde{g}_i^{(t)}}{\sum_i \tilde{N}_i^{(t)}} \right) \right)$

28:     **Buffer**$_i \leftarrow$ `All_Gather`$\left( \text{Shard}_i \left( \tilde{\theta}^{(t+1)} \right) \right)$

29:     $N_i^{(t)} \leftarrow 0$

30:     `Ready_for_Stage_2` $\leftarrow$ **True**

31:     `Ready_for_Stage_1` $\leftarrow$ **False**

32:     **Stage 2.**

33:     $(N_i^{(t)}, g_i^{(t)}) \leftarrow$ **Buffer**$_i$

34:     $\sum_i N_i^{(t)} + \tilde{N}_i^{(t)} \leftarrow$ `All_Reduce`$(N_i^{(t)} + \sum_i \tilde{N}_i^{(t)})$

35:     `Shard`$_i \left( \sum_i g_i^{(t)} + \tilde{g}_i^{(t)} \right) \leftarrow$ `Reduce_Scatter`$(g_i^{(t)} + \sum_i \tilde{g}_i^{(t)})$

36:     `Shard`$_i \left( \theta^{(t+1)} \right) \leftarrow$ `ShardedOpt` $\left( \text{Shard}_i \left( \theta^{(t)} \right), \text{Shard}_i \left( \frac{\sum_i g_i^{(t)} + \tilde{g}_i^{(t)}}{\sum_i N_i^{(t)} + \tilde{N}_i^{(t)}} \right) \right)$

37:     **Buffer**$_i \leftarrow$ `All_Gather`$\left( \text{Shard}_i \left( \theta^{(t+1)} \right) \right)$

38:     $\tilde{N}_i^{(t)} \leftarrow 0$

39:     `Ready_for_Stage_1` $\leftarrow$ **True**

40:     `Ready_for_Stage_2` $\leftarrow$ **False**

---

