# OpenReview forum: "ACCO: Accumulate while you Communicate, Hiding Communications in Distributed LLM Training"
_ICLR.cc/2025/Conference — Submitted to ICLR 2025_

### Official Review · Reviewer_by2S · 2024-10-28

**Soundness:** 2
**Presentation:** 3
**Contribution:** 2
**Rating:** 5
**Confidence:** 4

**Summary:**

This paper proposes ACCO, a memory-efficient optimization algorithm tailored for distributed training of LLMs. ACCO introduces 1-step delay to overlap the execution of gradient computations and communications, and then compensate this delay via weight prediction. The experiments show that in both pretraining and finetuning tasks, ACCO can accelerate training almost the same accuracy compared to the baseline.

**Strengths:**

1. This paper proposes ACCO, a memory-efficient optimization algorithm tailored for distributed training of LLMs. ACCO introduces 1-step delay to overlap the execution of gradient computations and communications, and then compensate this delay via weight prediction. The experiments show that in both pretraining and finetuning tasks, ACCO can accelerate training almost the same accuracy compared to the baseline.
2. The proposed algorithm is compatible with data-parallel sharding.
3. The paper also proposes a strategy to maximize GPU usage, even with heterogeneous hardware.

**Weaknesses:**

My major concern is that some part of the proposed algorithm has already been proposed in some previous papers, which are not discussed. Regardless of some systematic improvement (adapting to DP sharding and heterogeneous hardware), 1-step delay and delay compensation would weaken novelty and contribution, and probably make the contribution incremental compared to those previous work. For more details:

1. SGD with one-step delay (DPU explained at the beginning of Section 3) is actually PipeSGD [1]. However PipeSGD is not cited or discussed.

2. PipeSGD + delay compensation has been proposed in SAPipe [2]. However SAPipe is not cited or discussed. And in SAPipe paper, this compensation method is also called weight prediction (WP). Although there are some differences in details, the delay compensation proposed in ACCO can generally be viewed as a variant of SAPipe, additional to the 3 options already proposed in SAPipe. To the best of my knowledge, Option 1 of SAPipe is the one closest to ACCO, where SAPipe Option 1 uses latest local gradients without synchronization, and ACCO uses the latest but half-accumulated local gradients with synchronization applied.


--------------------------
References
[1] Li, Youjie, et al. "Pipe-SGD: A decentralized pipelined SGD framework for distributed deep net training." NeurIPS 2018.
[2] Chen, Yangrui, et al. "SAPipe: Staleness-Aware Pipeline for Data-Parallel DNNTraining." NeurIPS 2022.

**Questions:**

1. The weight prediction of ACCO uses half-accumulated gradients (1st stage of gradient accumulation). Is it because the 2nd stage of gradient accumulation is required to overlap with and hide the communication (synchronization, averaging across workers) overhead of these half-accumulated gradients? If so, what about using the locally accumulated gradient without averaging across workers (I think it is still compatible with data-parallel parameter sharding), which will allow the algorithm to use more than half or even the entire accumulated gradients? Basically, I think there could be a tunable trade-off between accumulation error (difference between half-accumulated gradients and fully-accumulated gradients) and synchronization error (difference between local gradients and globally synchronized gradients).

2. What about theoretical analysis of ACCO? I see there is a short discussion at the end of Section 3 but no rigorous analysis/proof is provided for convergence. I think the theoretical analysis from SAPipe paper could be easily modified to adapt to the convergence analysis of ACCO.

---

> ### Author Response · Authors · 2024-11-21
>
> We thank reviewer by2S for their thorough analysis of our paper, for acknowledging the strengths of our method, and for the suggested literature.
>
> **W1:**
>  >”PipeSGD is not cited or discussed.”
>
> We thank the reviewer for this relevant citation that we missed, and will add it to our related work section in the next iteration of our paper. We will also make clear in the different parts of our paper that “DPU” is not the only name of this particular method, but that it also appeared in other papers under other names.
>
> **W2:**
>  >”PipeSGD + delay compensation has been proposed in SAPipe”
>
> We thank again reviewer by2S for this highly relevant citation! We will make sure to discuss it in the next version of our paper. SAPipe introduces 3 strategies (see section 3.2 of SAPipe):
> 1) WP with local gradient in the current step,
> 2) WP with the latest synchronized gradient,
> 3) WP with all the above combined and the delay compensation technique from [[Zheng et al., 2017]](https://arxiv.org/pdf/1609.08326 ).
>
> However, if we partition the optimizer’s states, i.e. split the All-Reduce operation in two parts All-Reduce = Reduce-Scatter + All-Gather (see [NCCL doc](https://docs.nvidia.com/deeplearning/nccl/user-guide/docs/usage/collectives.html#allreduce ) for visuals on collective communication primitives), methods (1) and (3) **are not** implementable, which is a major drawback to train LLMs due to memory constraints (see line 60).
>
> Indeed, see in Fig.1 that the “Opt step” is sandwiched between the two communication parts Reduce-Scatter and All-Gather in the communication stream, which is run **in parallel** to the computation stream. Thus, when the Opt. step is done, only **half** of the communication load is finished, meaning that the computation is not finished either (the computation continues until the communication finishes) and the “local gradient in the current step” is thus not available because only half of it has been computed so far. This means that methods (1) and (3) cannot be implemented with sharded optimizers as they require access to the current gradient in the optimizer update. Moreover, sharding the optimizer leads to an update that can only be made in in conjunction with a communication: line 11 in Algorithm 3 of SAPipe cannot be done locally but only while communicating.
>
> However, method (2) re-uses the last averaged gradient to make a prediction, which is exactly the method we experimented with under the name “ACCO-WP” (and is inspired by [[Chen et al., 2019]](https://arxiv.org/pdf/1809.02839 ) ) - however, we show in our ablations line 391 that ACCO completely outperforms ACCO-WP thanks to the novel interleaved update method we introduce. We will add the reference to SAPipe when talking about ACCO-WP in the final paper, and how ACCO allows us to overcome its limitations.
>
> **Questions:**
>
> 1) The main reason for this interleaved update method using half-accumulated gradients is to make the method compatible with sharded optimizers, as the algorithms using locally accumulated gradients are not compatible with this (see the remark above on SAPipe). Thus the idea of *“using the locally accumulated gradient without averaging across workers”* is impossible in practice with sharded optimizers.
> 2) We agree that our theoretical analysis is not the most precise as it is, and the ones in SAPipe could help us in this regard. However, as the theoretical conclusions for standard 1-step delayed SGD for smooth functions do not seem to apply in practice for using AdamW to train Transformers (see our experiments with DPU), we preferred to rather focus on practical and empirical results for our paper.

---

> > ### Comment · Reviewer_by2S · 2024-11-22
> > **thanks for the response**
> >
> > Thanks for your insightful response
> >
> > I agree that strategy (3) of SAPipe may not be easily implemented in DP sharding.
> >
> > However, for strategy (1) I think it's totally implementable. For DP sharding, typically there is some kind of range mapping  between the sharded main parameters (the fp32 version for optimizer) and the local model parameters (used for forward-backward), this mapping is typically used for parameter all-gather, but we could also use it (or establish another mapping for the local gradient) to map the positions between the local gradient (before gradient synchronization) and the main gradient (the sharded one after reduce-scatter is done). Then, by using this mapping, we could slice the local gradient into the part that matches the local shard of the main parameters, and then do the weight prediction with it. There won't be extra memory footprint compared to strategy (2).
> >
> > Well... I'm not sure whether strategy (3) of SAPipe could also be implemented in this way, but it's possible I think.

---

> ### Author Response · Authors · 2024-11-23
>
> We thank reviewer by2S for their suggestion.
>
> We agree that it is possible to slice *a* local gradient vector and use it in the opt step.
> But we would also remark that our goal is to run computations in parallel to communication *and* use a sharded optimizer. Whatever strategy we may want to use, this means that the opt. step is done in the communication stream after the reduce-scatter and before the parameter all-gather while some gradient is computed in the computation stream. Thus, during the opt. step at time $t$, we have access to the following information:
> * The gradient computed locally at the last time step $t-1$ that we can slice.
> * The averaged gradient (after reduce-scatter) computed at the last time step $t-1$.
>
> But we do **not** have access to the gradient that we are *still* computing locally at time $t$ because this one will finish to be computed *after* the all-gather operation that is done *after* the opt. step.
> Thus, strategies (1) and (3) of SAPipe are not implementable.
>
> If the optimizer states were not sharded, then the opt. step could be done in the computation stream right after a gradient computation and the SAPipe strategies could be implemented, but this is not the case here.

---

> > ### Comment · Reviewer_by2S · 2024-11-23
> > **need more clarification**
> >
> > I actually cannot understand what do you mean by "we do not have access to the gradient that we are still computing locally at time because this one will finish to be computed after the all-gather operation that is done after the opt. step"
> > The "all-gather" you mentioned above is the parameter all-gather, right? If so, then how could the computation of the local gradient be finished after the parameter all-gather? The order must be: finish computing local gradient -> gradient reduce-scatter -> opt. step -> parameter all-gather. If the computation of local gradients is finished after the parameter all-gather, how could the gradient reduce-scatter and opt. step be done?
> >
> > Also, in your own algorithm, you do have access to the local gradient (the first-half mini-batch) for the reduce-scatter and then the opt. step for the weight prediction. But now you are telling me that without the reduce-scatter operation, you suddenly lose the access to the local gradient? I really don't understand this.
> >
> > Probably the authors misunderstood what I was suggesting. I'm basically saying that strategy (1) of SAPipe could be viewed as  ACCO but without half-gradient reduce-scatter in the weight prediction. I mean, I don't really think using half-minibatch local-gradient instead of full local-gradient in strategy (1) of SAPipe makes a big difference, since theoretically it's just another unbiased gradient estimator (of the local gradient) with larger variance.

---

> ### Author Response · Authors · 2024-11-28
>
> We thank reviewer by2S for precising their concerns, and hope to clear our explanations in the following.
>
> > *"how could the computation of the local gradient be finished after the parameter all-gather?"*.
>
> When training Transformers with a sharded optimizer, we have two versions of the parameter vectors $x$ and $x_{\text{slice } i}$ on each worker, one solely to *compute gradients* in half precision, the other to *update the model parameters* in full precision:
> * $x$ is the full vector of model parameters $\in\mathbb R^d$, stored in half precision (brain float 16) used to *compute gradients*.
> * $x_{\text{slice } i}$ is a full precision version (float32), but part of the optimizer's state, i.e a $1/N$ slice $\in \mathbb R^{d/N}$ (we assume that $d$ is divisible by $N$ for simplicity) used to *update the parameters*.
>
> If we have a communication/computation overlap *and* use a sharded optimizer, then we have 2 parallel processes at step $t$ on worker $i$:
> * $\nabla$ **computation**: while the communication is not finished, use $x_t$ to compute gradients $g_i^{(t)}$.
> * **communication**: Reduce-Scatter the gradients computed at last step $( \frac 1 N \sum_i g_i^{(t-1)} )\_{\text{slice } i}$ $\rightarrow$ use the sharded optimizer to update the $1/N$ slice $x_{\text{slice } i}^{(t)}$ with an Opt. step and thus create $x_{\text{slice } i}^{(t+1)}$ $\rightarrow$ All-Gather a half-precision version of parameters $x_{\text{slice } i}^{(t+1)}$ to create the full parameters (in bf16) $x_{t+1}$ for the next step.
>
> We hope that our explanation makes clear how the computation of local gradients is finished after the parameter all-gather.
>
> >*"you suddenly lose the access to the local gradient?"*
>
> We must emphasize that the optimizer being sharded (local worker $i$ has only access to a $1/N$ slice of the optimizer states), an optimizer step can only be made in conjunction with a communication. Without a communication, given a local gradient $g_i \in \mathbb R^d$, each worker $i$ could only update a slice of size $\mathbb R^{d/N}$.
>
> What we are saying is that for its weight prediction methods (1) and (3), SAPipe requires an update spanning over the *whole local model* $\in \mathbb R^d$ using local gradients $g_i \in \mathbb R^d$ but **without** communications (line 11 of Algo.3 of the SAPipe paper). This is not possible with sharded optimizers, as the local optimizer shard can only update a slice in $\mathbb R^{d/N}$.
>
> Note that ACCO only uses gradients *averaged* over the whole $N$ workers in its parameter updates and weight predictions (equations (1)-(2)), meaning that the update and weight predictions are done with a communication, allowing updates spanning the whole model $\in \mathbb R^d$.

---

### Official Review · Reviewer_j9YN · 2024-11-03

**Soundness:** 3
**Presentation:** 3
**Contribution:** 3
**Rating:** 6
**Confidence:** 3

**Summary:**

The authors propose The ACCumulate while COmmunicate (ACCO) algorithm, which is a memory-efficient distributed optimizer. It overlaps gradient computation and communication to reduce overhead and enables sharding optimizer states. Experiments show ACCO achieves communication and memory efficiency in large language model training and fine-tuning.

**Strengths:**

1. Figure 2 is a good illustration of the proposed method where the authors propose an interleaved update method. The idea appears very novel.
2. The proposed method can be adopted in ZeRO, without obvious memory overheads.
3. The proposed method achieve similar convergence curves as DDP but higher throughput.
4. Memory efficiency as summarized in Table 1. Though it's still not as efficient as ZeRO1.

**Weaknesses:**

1. From Table 2, the performance of ACCO deteriorates when increasing the number of workers from 8 to 32. There might be an scaling issue for the proposed method. Could you provide model performance v.s. number of workers like Figure 3?
2. Lacking in theoretical analysis. It appears easy to have a convergence analysis given the update formulation. It will be better to see how $\tilde{\theta}$ affects the convergence.
3. It does not seem appropriate to claim "as a form of plain SGD with no delay", because part of the gradient is computed at $\tilde{\theta}$ instead of $\theta$.

**Questions:**

1. Do you update optimizer states in both Eq(1) and (2), or only Eq(2)?

---

> ### Author Response · Authors · 2024-11-21
>
> We thank reviewer j9YN for their kind comments, and are happy that they find our method very novel and recognize its strengths.
>
> >”Could you provide model performance v.s. number of workers like Figure 3?”
>
> In Tab.2, the pre-training experiments with 8 and 32 GPUs were done on different hardware (see line 421): the 8 GPUs experiment was performed on a single node of H100, and the 32 GPUs one was on the A100 cluster described line 307.  Moreover, due to hardware availability constraints, we stopped the experiments on the 8 GPUs sooner than the ones on 32 GPUs (see the #tokens entry in Tab.3). So we are not able to fairly compare the two to this date. However, given more time (to secure the access to the necessary hardware for the corresponding length of time) we can re-run our experiments on 8 and 16 GPUs on our A100 cluster for the same number of tokens as our 32 GPUs experiment and display the results as in Fig.3. We will provide this experiment in our final version of the paper.
>
>
> >”Lacking in theoretical analysis. It will be better to see how  $\tilde \theta$ affects the convergence.”
>
> We agree that our theoretical analysis is not the most precise as it is. However, due to the interleaved update of ACCO, our attempted derivations do not appear to be that straightforward compared to classical “delayed SGD methods” either. Moreover, as the theoretical conclusions for standard 1-step delayed SGD for smooth-convex functions do not seem to apply in practice for using AdamW to train Transformers (see our experiments with DPU), we preferred to rather focus on practical and empirical results for our paper.
>
> >”It does not seem appropriate to claim "as a form of plain SGD with no delay"”
>
> We agree with the reviewer on that point and will re-formulate the sentence in the next version of our paper.
>
> >”Do you update optimizer states in both Eq(1) and (2), or only Eq(2)?”
>
> This is a great question, thank you! We only update the optimizer state for Eq (2). In our implementation, for Eq (1), we checkpoint the optimizer states right before taking the optimizer step and reverse to them right after, so that they are only updated for Eq (2).

---

### Official Review · Reviewer_tJ8M · 2024-11-03

**Soundness:** 2
**Presentation:** 2
**Contribution:** 2
**Rating:** 5
**Confidence:** 3

**Summary:**

The paper proposed an algorithm named ACCO, which jointly optimizes the memory cost and communication in LLM training. ACCO reduces communication overhead by concealing it under gradient computing. Experiments have been done on some pre-training tasks and fine-tuning tasks, which prove the effectiveness of ACCO.

**Strengths:**

•	Algorithm Design: The paper provides a new algorithm to conceal communication overhead of distributed training under gradient computing to support communication-efficient parallel training strategy.

•	Empirical Validation: The method is verified on some pre-training and fine-tuning tasks, results show that in some condition the time cost of distributed LLM training has been reduced a lot.

**Weaknesses:**

•	Experiment Weakness: The experiment is not rigorous enough, and it does not show a clear advantage when compared with the baseline DDP. The chosen pre-training and fine-tuning tasks are also not broad and sufficient.

•	Motivation Weakness: While the paper aims to optimize the distributed training process of large language models (LLMs), it would benefit from providing deeper insights and observations that clarify why the ACCO algorithm is particularly advantageous for LLMs, how split the mini-batch would influence the convergence. This additional context would strengthen the rationale behind the proposed approach.

•	Contribution to Memory Optimization is Weak: Although the paper claims to address both memory and communication challenges, its primary contribution lies in proposing a pipeline that divides the computation of mini-batch gradients into two successive stages. This approach primarily focuses on mitigating communication overhead rather than optimizing memory usage. Memory cost is more than ZeRO-1 actually.

**Questions:**

Looking at weakness part.

•	Add more experiments ?

•	Could you provide more insights and observations on why the ACCO algorithm is particularly beneficial for training large language models (LLMs)?

•	Are there specific cases or experimental results that support the unique advantages of the ACCO algorithm for LLMs?

•	Beyond mitigating communication overhead, does your approach offer direct contributions to memory optimization?

---

> ### Author Response · Authors · 2024-11-21
>
> We thank reviewer tJ8M for their questions, and hope to clarify our contributions in the following.
> >”why the ACCO algorithm is particularly beneficial for training large language models (LLMs)?”
>
> LLM training is particularly challenging because of the number of parameters  in the model (typically in the tens of billions or more). This poses two challenges for distributed training with data parallel:
> 1) the GPU memories are maxed out before fully hosting the models and optimizers states copies (see the memory footprints in our Tab.1),
> 2) the communication load necessary to average the large gradient vectors over the distributed workers at each step can be so large that it takes more time to perform than actually computing them (see our Fig.3).
>
> Thus, for LLM training, due to the models’ sizes and from a memory footprint viewpoint, it is **necessary** to use a sharding method (see line 60). But problem (2) is not tackled by partitioning methods such as ZeRO-1, leading to GPUs remaining idle for most of the time due to communication bottlenecks (see our Fig.3). ACCO is the first method allowing to tackle both problems (1) and (2) effectively, leading to significant *time* speedups (see Tab.3) compared to methods only tackling (1).
>
> >” Are there specific cases or experimental results that support the unique advantages of the ACCO algorithm for LLMs?”
>
> Yes, in the two main LLM training tasks (LLMs pretraining and LLM finetuning), we showed in Tab.3  that ACCO allows to train LLM models as good as standard DDP with ZeRO methods, with a similar training dynamic and memory footprint, but significantly faster thanks to the parallel execution of computations and communications.
>
> >”does your approach offer direct contributions to memory optimization?”
>
> We do not claim that ACCO introduces a new memory splitting method (it is the same as in ZeRO-1). However, ACCO is the first method that does **both** memory splitting and parallel execution of communication and computations, which is not trivial (as shown with our ablations and experiments with DPU). As shown in Fig.3 and Tab.3, memory optimization methods such as ZeRO are greatly impacted by communication overheads. On the other hand, previous methods parallelizing computations and communications such as CO2 cannot be seriously considered for LLMs training at scale due to the severe memory overhead they incur (see Tab.1). ACCO efficiently solves the two separate problems with a new approach.
>
> >”Memory cost is more than ZeRO-1 actually.”
>
> This is indeed the case, but a natural by-product of the parallel execution of gradient computations and averaging: the two processes editing a “gradient” vector, they must refer to different memory slots. Thus **all** methods parallelizing computations and communications need an additional communication buffer, and ACCO is not different in that regard. However, the buffer being in half precision (see our Tab.1 and  Fig.4 for an illustration), the memory overhead compared to ZeRO-1 is arguably negligible compared to the significant time speedups that it brings.

---

> > ### Comment · Reviewer_tJ8M · 2024-11-25
> >
> > Thank you for your response, but my concerns remain, so I will keep my original score.

---

### Official Review · Reviewer_a1EP · 2024-11-04

**Soundness:** 3
**Presentation:** 2
**Contribution:** 3
**Rating:** 6
**Confidence:** 1

**Summary:**

This paper proposes a gradient accumulation technique to address the communication overhead in synchronous training while maintaining low memory consumption. The approach involves splitting each mini-batch into two parts: the first half is used to compute an approximate next step of the model parameters, while the full mini-batch is then used to calculate a precise gradient update. The authors validate the effectiveness of this method through large-scale experiments.

**Strengths:**

This paper addresses a highly relevant challenge in large-scale model training: addressing the communication overhead that arises in synchronous distributed training, especially in large-scale scenarios. This approach addresses this challenge while keeping memory consumption low. The experimental results presented in this work are compelling. The authors show notable reductions in training time on realistic large-scale scenarios while keeping performance levels comparable to a standard synchronous method.

**Weaknesses:**

* The gradient accumulation technique in this approach may introduce numerical stability issues, especially when working over long sessions, when slower devices are involved. These accumulated errors might degrade the quality of updates, potentially resulting in exploding gradients or suboptimal model performance.

* The paper would benefit from a clearer presentation. For example, it lacks clarity in explaining how sharded optimizers specifically handle memory-intensive optimizer parameters and their distribution across workers. A more detailed description of sharded optimizer mechanics would enhance the paper’s accessibility and improve readers’ ability to fully capture the memory reduction benefits of the proposed approach and its advantage over local updates.

**Questions:**

Which optimizer parameters contribute to high memory consumption? How do sharded optimizers work to reduce this memory load? Specifically, which optimizer parameters are divided among the workers, and what information is required for a sharded local optimization step, such as the one used to compute the estimated model parameters in this approach?

---

> ### Author Response · Authors · 2024-11-21
>
> We thank reviewer a1EP for finding our method highly relevant and our experiments compelling.
>
>
> **Weaknesses:**
>
>
> * *Numerical stability issues when slower devices are involved:* first, we must emphasize that ACCO does not use "locally computed minibatch gradients" in its optimizer updates but rather *averaged* gradients over all workers (see Fig.1 and equations (1)-(2)). Thus if one worker is slow and do not have time to accumulate for multiple steps to reach a large batch-size locally, it does not affect the stability of the training procedure as every worker has access to the averaged gradient when updating parameters. However, we agree that the presence of arbitrarily slow workers can negatively impact the training speed for our method. In a typical data center, GPUs can crash for multiple reasons when training LLMs (see the [Llama 3 paper]( https://arxiv.org/pdf/2407.21783 )). Our method does not handle this extreme case, and akin to what is done today when this happens, the solution is to remove the faulty GPU from the pool as it would only slow down the training.
> * *Explanation of sharding optimizers:* we agree that a more detailed explanation of the sharding optimization would help clarify the paper for readers not familiar with this literature and thank the reviewer for this comment. We will add a paragraph detailing this in the final version of our paper.
>
>
> **Question:**
>
>
> The sharding method we follow is the same as [ZeRO-1](https://arxiv.org/pdf/1910.02054 ), but contrary to previous literature, ACCO allows to shard optimizer’s states across workers while running computation and communications in parallel and implementing an effective delay compensation method.
>
> In a typical data parallel method for LLM training, each distributed worker needs to host in its local memory a copy of the model’s parameter in bf16, a gradient buffer in bf16, and for the optimizer: a float32 version of the parameters, and two float32 moment vectors. Taking the notations of our Tab.1, this means that for a model with $\Phi$ parameters, each of the $N$ workers hosts $2\times 2 \Phi$ Bytes for the model and gradient buffer, and $4 \times 3 \Phi$ Bytes for the optimizer states. As shown in our figure 4, here the sharding strategy allows each distributed worker to only host $\frac{4 \times 3} N \Phi$ Bytes for the optimizer states (the bf16 parts are untouched). This is done by splitting the All-Reduce communication primitive into two halves: Reduce-Scatter and All-Gather. In standard DDP, each worker computes its gradients, they are then averaged through all-reduce and each worker has thus access to the full averaged gradient before taking an optimizer step over the full $\Phi$ parameters. Here, each worker stores only $1/N$ of the optimizer states, and can thus only update a slice of the parameters. But as All-Reduce = Reduce-Scatter + All-Gather *(see [NCCL doc](https://docs.nvidia.com/deeplearning/nccl/user-guide/docs/usage/collectives.html#allreduce ) for visuals on collective communication primitives)*, instead of doing All-Reduce $\rightarrow$ Opt step, one can do Reduce-Scatter $\rightarrow$ Opt step over the $1/N$ slice $\rightarrow$ All-Gather for the exact same result and communication cost, but with a reduced memory cost, which we do (see our Fig.1).
>
> In practice in our code, we just split the optimizer states in $N$ equal slices, each distributed worker only hosting its slice (which requires knowing the world size and the rank of the worker at initialization, which is common for distributed methods).

---

### Official Review · Reviewer_aeVa · 2024-11-06

**Soundness:** 2
**Presentation:** 2
**Contribution:** 1
**Rating:** 3
**Confidence:** 4

**Summary:**

The submission presents a method for training large models in a distributed setting where the computation and communication overlap. The method has been named "ACcumulate while COmmunicate or ACCO". The work extends the delayed parameter updates of sharded training to introduce an estimate of the model state after the step when the current ongoing gradient computation would have been applied.

The introduced algorithm has been evaluated on pre-training of GPT-neo 36M on TinyStories dataset, GPT-neo 125M on OpenWebText dataset, and fine-tuning of GPT-neo 2.7B on Alpaca dataset.

The results claim that the presented method has better time to accuracy in some cases, such as for finetuning over 4 nodes.

**Strengths:**

The submission attempts to address a significant problem in contemporary machine learning, which is improving the scalability and efficiency of training algorithms for large language models. The presentation is fair and highlights the performance of the proposed method.

**Weaknesses:**

The submission lacks both novelty and insights. More specifically,
* The delayed update with estimated gradients is a well-known technique and has already been even theoretically analyzed for its convergence in distributed machine learning; see "Elastic consistency: A practical consistency model for distributed stochastic gradient descent. Nadiradze et al. AAAI 2021, Section: Elastic Scheduling". In essence, introducing estimated gradients before the actual update while gradient updates are computed for the previous batch is well-known and implemented. Elastic scheduling of Nadiradze et al. also showed better time to accuracy for the model it trained.
* The theoretical discussion of the work is only at a very high sketch level. Again, Elastic consistency for the convergence of the presented algorithm could be a good approach here.
* There is no insight into why the performance gain happens in some cases and not in multiple other cases. In this context, it is less than fair to advertise this work as "Comparedto ZeRO, our implementation and experiments on several LLMs pre-training and fine-tuning tasks demonstrates that ACCO reduces the learning time upto 87% and successfully allows both sharding optimizer states across workers and the use of heterogeneous hardware."

**Questions:**

1. Can you please comment on how ACCO differs from Elastic Scheduling? Sharding is a system-related aspect brought into elastic scheduling. However, this is only a very incremental work described in some detail in the submission. Sharding in its merit is not novel; standard SGD implementation to achieve a minibatch size that does not fit on GPU memory naturally requires sharding.
2. Can you please explain why ACCO does not outperform DDP in Figure 6 results?

---

> ### Author Response · Authors · 2024-11-21
>
> We thank reviewer aeVa for the suggested reference, we will make sure to add it to our literature review as it is relevant to our work. The “[Elastic Scheduling]( https://arxiv.org/pdf/2001.05918)” paper introduces a theoretical framework to analyze the convergence property of a relaxed version of distributed SGD, showing that *some* degree of computation/communication overlap can theoretically be possible for smooth functions, and experiments with a 20\% computation/communication overlap with ResNet28 on CIFAR-100.
>
>
> However, we respectfully disagree that our method is incremental compared to it : Elastic Scheduling requires 80% of a non-delayed gradient, making the computations and communications sequential for the most part whereas ACCO allows a perfect overlap of gradient computations and communications, which by design is a major difference and a huge advantage in terms of speed. By contrast, the purposes of ACCO are three-fold:
> 1) Running gradient computations and communications in parallel, hiding 100\% of the communication costs (as opposed to some).
> 2) Allowing sharding the optimizer states.
> 3) Matching the training dynamic of standard backpropagation when training LLMs with the AdamW optimizer.
>
> We believe that taken together and in consideration that our method is feasible for LLM  is a significant step: “Elastic Scheduling” only partially address (1) *(but does not achieve full overlap)* for SGD with ResNets. However, ACCO is the first method perfectly meeting the three goals together and evaluated on LLMs.
>
> We emphasize that prior work has shown evidence  that delayed methods work fine in theory for SGD on smooth functions, and in practice in the case of using SGD for training ResNets (cf e.g. “Elastic Scheduling”). However, we demonstrate empirically that for AdamW on Transformers, this is not the case (cf our experiments with DPU). This calls for mitigating methods. Given the nonlinear update rule of AdamW and the regularity of Transformers’ loss function, our purpose is not to provide a theoretical answer (that would be harder to derive than the analysis for “SGD with smooth functions”), but rather a practical method that is demonstrated empirically. We show in our ablation (line 392) that not all mitigating methods are equally good at compensating delays in this context, but ACCO does it perfectly.
>
> While “Elastic Scheduling” provides a theoretical analysis for SGD with smooth functions, it does not experiment with training Transformers with AdamW, only hides 20\% of the communication cost *($\beta=0.8$, use of the $L0$ norm in their experiments page 8)*, and does not try to implement (2), which is not straightforward nor possible for all methods designed for (1) and (3) (see line 123, or our answer to reviewer by2S). ACCO does all that.
>
>
> >”There is no insight into why the performance gain happens in some cases and not in multiple other cases”
>
> Would it be possible to point us to the precise inconsistencies you see to help us clarify our paper please? We demonstrated in Fig.3 that communication is a bottleneck, our goal is to design a method matching the training loss and memory requirement of standard backpropagation (DDP with a sharded optimizer) while being faster. In terms of loss/sample seen, all our Figures show that DDP and ACCO behave in the same way. The difference in the time gains reported in Tab.3 depend on the size of the model, the number of GPU nodes, and the communication bandwidth between the GPUs: all of these variables make the communication bottleneck more or less severe and are reported in Tab.3.
>
> >”Can you please explain why ACCO does not outperform DDP in Figure 6 results?”
>
> Figure 6 shows the evolution of the training loss with respect to the number of training samples for 3 methods: DDP (baseline), ACCO-wp (ablation) and ACCO (our final method). The results show that ACCO-wp is not as good as our final method, ACCO, at  countering the negative effect of the delayed gradient. The ACCO curve (blue cross) perfectly overlaps the DDP curve (plain purple), demonstrating that it perfectly compensates the delay. Tab.3 shows that it outperforms DDP in terms of training time.

---

> > ### Comment · Reviewer_aeVa · 2024-11-26
> > **Rebuttal to review**
> >
> > Thanks for your response.
> >
> > >that would be harder to derive than the analysis for “SGD with smooth functions”
> >
> > Yes. There are other existing works that derive it for non-smooth objectives: "Asynchronous SGD Beats Minibatch SGD
> > Under Arbitrary Delays, Mishchenko et al." So, this is not an open question/challenge.
> >
> > > While “Elastic Scheduling” provides a theoretical analysis for SGD with smooth functions, it does not experiment with training Transformers with AdamW.
> >
> >
> > Are you suggesting that experiments with AdamW is a contribution?
> >
> > My concerns still remain the same: Sharding in itself is not a novelty, and computation-communication overlap is not a novelty; could you please list your novelty in a table in terms of design and discovered insights other than doing experiments with some algorithms using known and widely adopted techniques of sharding and communication hiding?
> >
> >
> > For now my score is unchanged.

---

> ### Author Response · Authors · 2024-11-28
>
> We thank reviewer aeVa for their answer, and hope to completely address their concerns in the following.
>
> >There are other existing works that derive it for non-smooth objectives
>
> We must precise that we are talking about the theoretical study of optimizers with *non-linear update rules such as Adam* (not SGD) on less regular functions than the classical setting of "convex-smooth" that is far from truthfully depicting the loss landscape of Transformers. This is still an *open area of research*, see e.g. the conclusion of "[Advances in Asynchronous Parallel and Distributed Optimization, Assran et al.]( https://ieeexplore.ieee.org/document/9217472 )"
>
> > Are you suggesting that experiments with AdamW is a contribution?
>
> Not exactly: we are affirming that algorithms using a delayed update may work easily with SGD on convolutional neural networks in practice **but not** when using Adam on Transformers. See for example [Fig.4 of the SAPipe paper](https://proceedings.neurips.cc/paper_files/paper/2022/file/725ce5f2b1a8e2e0ac66994e7fefe375-Paper-Conference.pdf ) where Pipe-SGD (other name for the "1-step delayed update" or DPU) is shown to work perfectly well for SGD with VGG16 and ResNet50 on ImageNet/CIFAR10, but does not work at all for Adam on Transformers/GPT2. This proves there is a discrepancy between the optimistic conclusions of both the theoretical analysis of "delayed update methods for SGD on smooth functions" and simple experiments with SGD on CNNs, and what is observed in practice for Adam on Transformers.
>
> Our contribution is thus a offering a method that *does work* for AdamW and Transformers.
>
> > Sharding in itself is not a novelty, and computation-communication overlap is not a novelty
>
> We agree on this statement. However, having *both* at the same time **is** a novelty.
> We must insist that while sharding is straightforward to apply for standard backprogation, it is highly non-trivial to manage to design a method that allows to shard the optimizer's parameters in a "delay-compensating" framework (see our answer to reviewer by2S). The main reason for that is because many "weight prediction" methods use "local gradient approximates" in their update rule, which means that each worker must host a full optimizer and *not* a sharded one.
>
> We summarize in the following our contributions:
>
> | Algorithm          | 100\% Communication/Computation overlap | Allows sharding | Works for Adam on Transformers |
> |--------------------|-----------------------------------------|-----------------|--------------------------------|
> | Standard DDP       | No                                      | **Yes**         | **Yes**                        |
> | Elastic Scheduling | No                                | ?               | ?                             |
> | DPU/PipeSGD        | **Yes**                                 | **Yes**         | No                             |
> | SAPipe               | **Yes**                                 | No       | **Yes**                        |
> | ACCO               | **Yes**                                 | **Yes**         | **Yes**                        |

---

> > ### Comment · Reviewer_aeVa · 2024-12-03
> > **Response to the Authors**
> >
> > Dear Authors,
> >
> > Thank you for your response. This work looks better suitable for a system conference where the audience will be able to appreciate the technical contributions. To me, it looks like it is bringing no insights and no algorithmic/analytical improvement on the existing methods.
> >
> > I am not of the opinion of changing the score. The rest lies with the meta-reviewer.

---

> ### Author Response · Authors · 2024-12-03
>
> Dear reviewer aeVa,
>
> We thank you for your feedback and believe that your primary concern regarding the novelty/importance of our work may stem from some misunderstanding.While we agree that the working implementation of ACCO is a contribution in itself, we must emphasize that ACCO is a novel algorithm with a design justified through a comprehensive ablation study. ACCO is indeed motivated by performance issues (here, to tackle communication and memory bottlenecks when training LLMs) in the same fashion as many recent advances in optimization for deep learning (e.g. local SGD for the Federated Learning use case), but the solution we propose is an *algorithmic* one and not a mere implementation trick.
>
> Our novel algorithm vastly differs from previous ones in terms of design (see the unique interleaved update in Fig.2), and allows for implementation advantages impossible to obtain from previous methods (100% communication overlap, optimizer sharding).

---

### Author Response · Authors · 2024-12-03

We thank the reviewers for their comments and their questions.

The main concern raised by reviewers aeVA and by2S seems to be the comparison between ACCO and existing methods such as Elastic Scheduling and SAPipe. Contrary to those, ACCO is specifically designed to train LLMs (Large Transformers trained with AdamW), inducing specific challenges both in terms of (1) stability to delayed gradients and (2) memory, which are  **not** tackled by previous literature and are solved by ACCO.

1) As shown in our experiments with DPU (Fig.5 and Fig.8) as well as displayed in Fig.4 of the [SAPipe paper,]( https://proceedings.neurips.cc/paper_files/paper/2022/file/725ce5f2b1a8e2e0ac66994e7fefe375-Paper-Conference.pdf ) (Pipe SGD is another name for DPU) Transformers trained with Adam are particularly sensitive to the 1-step delay induced by parallel communication and computations of gradients, contrary to the optimistic conclusions of theoretical analysis of Delayed-SGD and experiments with SGD on ResNets (as done in Elastic Scheduling). This calls for specific *delay compensation methods*.

2) Due to memory constraints, training Large Models at scale **requires** the use of a partitionning strategy making sure that each distributed worker only holds $1/N^{th}$ of the optimizer's state. However, because each worker can only update $1/N^{th}$ of the model on its own using these sharding strategies, each model update **must** be done inside a communication step (to Reduce-Scatter and All-Gather). This restraints the nature of the delay compensation methods that can be used as *local* updates on the whole model are not possible anymore (e.g., SAPipe strategies).

As explained in our answers, ACCO is the *first and only* method to run parallel computations and communications when training LLMs *and* that allows to shard the optimizer's state while using an efficient delay compensation method, making it a significant contribution to the field compared to previous algorithms.

We hope this summary addresses the reviewers remaining concerns.

---

### Meta-Review · Area_Chair_axri · 2024-12-17

**Metareview:**

This paper proposes a method, ACCO, to improve the scalability and efficiency of distributed training for large machine-learning models. The approach leverages parameter sharding and overlapping communication and computation for base optimizers (e.g., SGD, AdamW).

The reviewers noted that overlapping communication and computation is a common technique in distributed learning and questioned the novelty of the idea. In response, the authors argued that ACCO achieves better overlap and scaling than prior methods.

While the reviewers acknowledged the significant engineering effort behind the implementation, they found the paper lacking insights relevant to a machine-learning audience. They also suggested that providing additional context to explain why ACCO is particularly advantageous for large language models (LLMs) would strengthen the rationale for the proposed approach.

Given these limitations, we recommend rejecting this version of the work.

**Additional Comments On Reviewer Discussion:**

The authors and the reviewers have discussed the work intensively. The discussion with reviewer aeVa clarified the work's contributions, and the discussion with reviewer by2S clarified its relation to prior work.

---

### Decision · Program_Chairs · 2025-01-22

Reject